Development of a new generation of miniemulsion based on cottonseed oil with α-tocopherol and ZnO and evaluation of its adjuvant activity

Sobrevilla-Hernández Gustavo 1
Franco-Molina Moisés Armides 1 moyfranco@gmail.com
Zárate-Triviño Diana G. 1
Kawas Jorge R. 2
Hernández-Martínez Sara Paola 3
García-Coronado Paola Leonor 1
Santana-Krímskaya Silvia Elena 1
Alvizo-Báez Cynthia Aracely 1
Rodríguez-Padilla Cristina 1
1 Laboratorio de Inmunología y Virología, Facultad de Ciencias Biológicas, Universidad Autónoma de Nuevo León , San Nicolás de los Garza, Nuevo León , Mexico
2 Posgrado Conjunto Agronomía-Veterinaría, Universidad Autónoma de Nuevo León , General Escobedo, Nuevo León , Mexico
3 Facultad de Agronomía, Universidad Autónoma de Nuevo León , General Escobedo, Nuevo León , Mexico
Gould Gwyn
Electronic publication date: 2023 Mar 20
Publication date: 2023
Volume: 11
Electronic Location ID: e14981
Received 2022 Oct 12; Accepted 2023 Feb 9
Copyright: © 2023 Sobrevilla-Hernández et al.
Copyright year: 2023
Copyright holder: Sobrevilla-Hernández et al.
License: This is an open access article distributed under the terms of the Creative Commons Attribution License, which permits unrestricted use, distribution, reproduction and adaptation in any medium and for any purpose provided that it is properly attributed. For attribution, the original author(s), title, publication source (PeerJ) and either DOI or URL of the article must be cited.
License URL: https://creativecommons.org/licenses/by/4.0/

Keywords: Immunoestimulant, ZnO nanoparticles, Cottonseed oil, Adjuvant, Emulsion

Funding: Fondo Sectorial de Investigación para la Educación, CONACYT, México A1-S-35951 Laboratorio de Inmunología y Virología Facultad de Ciencias Biológicas Universidad Autónoma de Nuevo León This research was funded by the Fondo Sectorial de Investigación para la Educación, (Grant Number A1-S-35951), CONACYT, México and the Laboratorio de Inmunología y Virología, Facultad de Ciencias Biológicas from the Universidad Autónoma de Nuevo León. The funders had no role in study design, data collection and analysis, decision to publish, or preparation of the manuscript.

==============================
Background

Emulsions have been widely used as immunological adjuvants. But the use of materials derived from plants such as cottonseed oil, alpha-tocopherol, or minerals such as zinc, as well as their use at the nanometric scale has been little explored. In this study, we develop a new miniemulsion and evaluated its antioxidant and phagocytic capacity, as well as parameters related to immune response stimulation by cytokine expression and antibodies production in a mice model.

Methods

Formulated CN (cottonseed oil miniemulsion) and CNZ (cottonseed oil miniemulsion whit zinc oxide nanoparticles) miniemulsions were characterized by scanning electronic microscopy SEM, DLS and FT-IR. In murine macrophages, splenocytes and thymocytes primary cultures safety and cytotoxicity were determined by MTT. In macrophages the antioxidant and phagocytic capacity was evaluated. In BALB/c mice, the stimulation of the immune system was determined by the expression of cytokines and the production of antibodies.

Results

The CN and CNZ presented stability for 90 days. Immediately after preparation, the CN presented a higher particle size (543.1 nm) than CNZ (320 nm). FT-IR demonstrated the correct nanoparticle synthesis by the absence of sulfate groups. CN and CNZ (1.25 to 10 µL/mL) had no toxic effect on macrophages (p = 0.108), splenocytes (p = 0.413), and thymocytes (p = 0.923). All CN and CNZ doses tested induced nitric oxide and antioxidants production in dose dependent manner when compared with control. CN-ovalbumin and CNZ-ovalbumin treatments in femoral subcutaneous tissue area showed inflammation with higher leukocyte infiltration compared with FCA. The intraperitoneal administration with CN, CNZ, and FCA showed a higher total intraperitoneal cells recruitment (CD14+) after 24 h of inoculation than control (p = 0.0001). CN and CNZ increased the phagocyte capacity with respect to untreated macrophages in the Candida albicans-phagocytosis assay. The evaluation of residual CFU indicated that only CN significantly decreased (p = 0.004) this value at 3 h. By other side, only CN increased (p = 0.002) the nitric oxide production. CNZ stimulated a major INFγ secretion compared with FCA at day 7. A major IL-2 secretion was observed at days 7 and 14, stimulated with CN and CNZ. Both miniemulsions did not affect the antibody isotypes production (IgG1, IgG2a, IgG3, IgA and IgM) at days 7, 14, 28, and 42. CN induced a significant IgG production against OVA, but lesser than FCA.

Conclusions

The two new miniemulsions with adjuvant and antioxidant capacity, were capable of generating leukocyte infiltration and increased cytokines and antibodies production.

Introduction

Adjuvants have been used to increase the antigens immunogenicity in vaccines or to modulate the immune system in diverse conditions of disease. Adjuvants can be obtained from plants, bacterial products, complex carbohydrates, animal extracts, and synthetic cytokines (Shahbazi & Bolhassani, 2016). Emulsion adjuvants employing mineral oils have an extensive clinical history use dating back to the 1930s (Shah et al., 2015). Different emulsion types have been used as immunostimulants or adjuvants in therapeutic or prophylactic vaccination, emulsions are formed when two immiscible liquids are brought together and one of them is organized into small droplets dispersed within the other, and stabilized by an interfacial surfactant layer. Emulsions can be formulated either as water-in-oil (W/O), oil-in-water (O/W) or water-in-oil-in-water (W/O/W) (Burakova et al., 2018). However, emulsions in some occasions can induce strong adverse reactions, such as local inflammatory lesions, pain, and distress, reason why, their use has been prohibited in animal and human vaccines (Burakova et al., 2018). Hence biocompatible oil-in-water emulsions based on vegetable or animal oils have been explored in efforts to improve tolerability, by decreasing the amount of oil and improving their biodegradability (Shah et al., 2015).

Squalene is a type of unsaponifiable lipid that acts as a biosynthetic precursor to all steroids in plants and animals (He & Corke, 2003). Some emulsions developed as adjuvants were based in squalene. Squalene-based adjuvants have been administered to humans, including children, and a good safety profile has been established. Given the public’s perception regarding to squalene safety, and misunderstandings regarding to its source, it is necessary to determine whether any other oil could be used as replacements for squalene (Fox et al., 2011). Vegetable oils can be used as an alternative adjuvant to squalene-based emulsions in vaccines and immunostimulants, because of their long variety of active substances that can affect the immune system. Several vegetable oils have α-tocopherol (Grilo et al., 2014). This one has immunomodulatory effect on dendritic cells (DCs), macrophages, natural killer (NK) cells, T cells, and B cells development, function and regulation (Lee & Han, 2018). Cottonseed oil contains α-tocopherol (411 ± 8.23 to 470.0 ± 9.4 mg/kg) (El-Mallah et al., 2011); AS03 commercial emulsion used as adjuvant in Prepandrix (pre-pandemic H5N1) and Pandemrix (pandemic H1N1) influenza vaccines contain α-tocopherol as immune potentiator (Shah et al., 2015). Otherwise, zinc is a nutritional mineral that participates in a variety of cellular functions, including signal transduction, transcription, and replication; strongly influencing the immune system affecting both nonspecific and acquired immunity (Dardenne, 2002). However, zinc oxide (ZnO) in nanoparticle form has not been used before as part of an immunogenic adjuvant. We aimed to develop a new miniemulsion and study its antioxidant and phagocytic capacity as well as parameters related to immune response stimulation by expression of cytokine and antibodies production in a mice model.

Materials and Methods

Minioemulsion development and characterization

Reactive

All reactive were used without additional purification, cottonseed oil pharmaceutical grade, α-tocopherol (96% purity), zinc sulfate monohydrate (ZnSO4 • H2O), sodium hydroxide (NaOH), Tween 20, and DMSO (dimethyl sulfoxide) were purchased from Sigma-Aldrich Inc (Saint Louis, MO, USA). Fetal bovine serum (FBS) (Thermo Fisher Scientific, Waltham, MA, USA), DMEM culture medium (Thermo Fisher Scientific, Waltham, MA, USA), 3-(4,5-dimethylthiazol-2-yl-2,5-diphenyltetrazol) bromide (MTT) (Thermo Fisher Scientific, Waltham, MA, USA), the Antioxidant assay kit (Catalog No. KA1622; Abnova, Taipei, Taiwan), Griess reaction (Griess Reagent, Catalog No. ab234044; Abcam, Cambridge, UK), Lipopolysaccharide (Sigma-Aldrich Inc., Saint Louis, MO, USA), Neutral Red Assay Kit-Cell Viability/Cytotoxicity (Catalog No. ab234039; Abcam, Cambridge, UK), Phagocytosis assay kit (IgG FITC) (Catalog No. 500290; Cayman chemical, Ann Arbor, MI, USA), C. albicans (ATCC 10261). Ovalbumin (OVA), (Sigma-Aldrich Inc., Saint Louis, MO, USA), Freund’s Adjuvant Complete (FCA), (Sigma-Aldrich Inc., Saint Louis, MO, USA). Ig Isotyping Mouse Uncoated ELISA kit (Catalog No. 88-50630; Thermo Fisher Scientific Inc., Waltham, MA, USA). Mouse Th1/Th2/Th17 Cytokine kit (Catalog No. 560485; DB Biosciences, San Jose, CA, USA). Formaldehyde (Catalog No. F8775; Sigma-Aldrich Inc., Saint Louis, MO, USA). Hematoxylin and eosin (H & E) (Catalog No. ab245880; Abcam, Cambridge, UK), CD14 Monoclonal Antibody (Sa2-8), APC, eBioscience (Catalog No.17-0141-82; Thermo Fisher Scientific Inc., Vienna, Austria).

Zinc oxide nanoparticles synthesis

The zinc oxide (ZnO) nanoparticles were obtained by the coprecipitation method. Zinc sulfate monohydrate (ZnSO4 • H2O) was dissolved in deionized water at 0.5 M that remained under stirring at 300 rpm while sodium hydroxide solution (2 M NaOH) was added in drops. After, the supernatant was discarded, and the precipitate formed was washed with deionized water and centrifuged at 8,000 rpm for 10 min. Precipitate was dried in a hot air oven at 60 °C for 24 h. Then, the obtained powder was calcinated at 300 °C for 2 h, in a muffle.

Miniemulsion formulation

Two miniemulsions were produced as follow: (1) cottonseed oil miniemulsion (CN) (to form the oil phase of 275 mg), in which α-tocopherol (25 mg), Tween 20 (35 mg) and DMSO (50 ml) were dissolved, and (2) cottonseed oil miniemulsion + ZnO nanoparticles (CNZ) was formulated in the same way as described above plus the addition of ZnO nanoparticles (2.5 µg) in the oil phase. The oil phase was calibrated with saline solution (1 mL) and mixed at 12,000 rpm in homogenizer for 10 min (ULTRA-TURRAX®; IKA Works Inc., Wilmington, NC, USA).

Miniemulsion characterization

Droplet size and polydispersity index (PDI) were determined by dynamic light scattering (DLS) and zeta potential (PZ) was determined by electrophoretic mobility measurements in a ZS90-Nano Zetasizer (Malvern Instruments, Malvern, UK). All results were measured in triplicate at room temperature (25 °C). The ZnO nanoparticles were characterized by X-ray diffraction (XRD) and the Fourier Transform Infrared Spectroscopy (FT-IR), ZnO nanoparticles morphology and size were confirmed by scanning electron microscopy (SEM), and the droplet size was confirmed by atomic force microscopy (AFM), images were analyzed with Gwyddion software. The characterization of miniemulsions functional groups and their components (cottonseed oil, α-tocopherol, and ZnO nanoparticles) were determinate by attenuated total reflection (ATR) in conjunction with FT-IR technique. The samples were placed onto the diamond ATR crystal, and spectra was recorded by a FT-IR (Perkin Elmer, Waltam, MA, USA), with an accumulation of 16 scans at a resolution of 4 cm−1 between 4,500–500 cm−1. Data were analyzed with the software Perkin Elmer Spectrum 10.

Animals

Male BALB/c mice (6–8 weeks old, weighing 23 ± 2 g) were used for all in vivo and ex vivo experiments. The animals were obtained from the Bioterium of Facultad de Ciencias Biológicas of the Universidad Autónoma de Nuevo León, (UANL, FCB) under the approval of Bioethics committee protocol No. CEIBA-2019-015. Animals were housed eight per cage (polypropylene) under constant specific conditions with temperature at 24 °C, 50% relative humidity and a controlled light and dark cycle (12 h:12 h) with ventilated caging systems, wood sawdust bedding and standardized environmental enrichment. Feed (pellet total mix ration) and clean water were supplied ad libitum. For sample (serum, macrophage, thymocyte and splenocyte) collection, animals were euthanized by overdoses of anesthesia (ketamine-xylazine protocol).

Cell obtention and culture conditions

Thymus, spleen, and peritoneal macrophages primary cultures were obtained from healthy mice. Cell cultures were maintained in a medium containing 5% fetal bovine serum (FBS), antibiotic and antimycotic 1%, under an atmosphere of 5% CO2 and 95% air at a constant temperature of 37 °C.

Ethical statement

All the experiments pertaining to animal use and their care strictly followed the guidelines of the Official Mexican Standard (NOM-062-ZOO-1999) on technical specifications for production, care and use of laboratory animals. All the protocols were approved by UANL FCB Animals Ethics Committee (protocol CEIBA-2019-015).

Cell toxicity

The miniemulsions toxicity was evaluated in vitro on thymus, spleen, and peritoneal macrophages primary cultures. Cells were seeded in 96-well plates at a density of 2 × 105 cells/well with 100 µL of DMEM culture medium. After 24 h of incubation culture medium was replaced by new one containing CN or CNZ (0, 1.25, 2.5, 5 and 10 µL/ml). Cell viability was evaluated 24 h after treatment using the technique of MTT.

The spectrophotometry measurement was performed in triplicate at 570 nm in a plate reader (Synergy HT; Biotec, Emmerich, Germany), the percentage of viability was determined by the following equation:

Viability(%)=(Absorbanceofthesample/Absorbanceofthecontrol)∗100

Macrophage NO (nitric oxide) and antioxidants production

For the miniemulsions effect over nitric oxide production (as nitrite) and total antioxidant capacity (TAC), murine peritoneal macrophages (MPM) were used. As positive control lipopolysaccharide (LPS) was used in a concertation of 5 µL/mL (Sigma-Aldrich Inc., St Louis, MO, USA).

Nitrite concentration in treated cells supernatant was determined by Griess reaction (Griess Reagent, Catalog No. ab234044; Abcam, Cambridge, UK). MPM were cultured in the same way as mentioned in the cell toxicity assays section described above. Supernatants (50 μL) were collected, centrifuged at 600× g for 5 min, mixed with 100 μL of Griess reagent (0.1% N-(1-naphthyl)-ethylene diamine, 1% sulfanilamide in 5% phosphoric acid) and incubated at room temperature for 10 min. Absorbance was measured at 540 nm (Bie et al., 2019).

TAC was determinate in MPM culture supernatants. MPM were cultured and supernatants were collected under conditions described above. The Antioxidant assay kit (Catalog No. KA1622; Abnova, Taipei, Taiwan) was used according to manufacturer’s instructions. Assay measures TAC by the reduction of Cu+2 to Cu+ observed by a colored complex formation. Trolox (6-hydroxyl-2,5,7,8-tetra methyl chroman-2-carboxylic) standard curve was measured at 540 nm.

Macrophage uptake and phagocytosis

Two assays were designed to determine the miniemulsions uptake and phagocytic activity effect in MPM. CN and CNZ (10 µL/mL) were used as cell stimulation treatment, for phagocytosis inhibition control disodium clodronate (0.6 mg/ mL) was used and DMEM medium as negative control. MPM quantitative uptake capacity was determinate by neutral red assay (Bie et al., 2019) using the Neutral Red Assay Kit-Cell Viability/Cytotoxicity (Catalog No. ab234039; Abcam, Cambridge, UK) according to manufacturer’s instructions. For the assay, 2 × 105 murine peritoneal macrophages were seeded on 96-well plates, washed three times with PBS, and then incubated with 10% neutral red for 2 h. After three washes, neutral red incorporated into lysosomes was dissolved by the lysis solution provided by the kit, and the absorbance was read at 540 nm. Qualitative phagocytic activity was determined using Phagocytosis assay kit (IgG FITC) (Catalog No. 500290; Cayman chemical, Ann Arbor, MI, USA). MPM were seeded on coverslips in six-well plates and incubated for 24 h in DMEM medium. After 1 h of treatments cells were incubated with latex beads-rabbit IgG-FITC complex for 2 h. Photographies were observed in fluorescence microscope (Axioscope 5 Pol; Zeiss, Oberkochen, Germany).

Candida Albicans macrophages phagocytosis

For the evaluation of the MPM phagocytic capacity, C. albicans (ATCC 10261) was used. MPM, obtained from three healthy mice were seeded under the previously described conditions in the cell toxicity assay. Phagocytosis assay was performed based in the method proposed by Alieke et al. (2002). Co-culture supernatant (macrophages (1 × 106 cells)/C. albicans (1 × 106–1 × 108 cells) proportion) was collected to assess the increase or decrease of colony forming units (CFU) by the Mc-Farland method (Vasconcelos et al., 2014). The percentage of phagocytized microorganisms was defined as following equation:

(1−(Numberofun−phagocytedCFU/CFUatthestartofincubation))∗100.

Death of C. albicans by phagocytes was assessed in the same monolayers. The nitic oxide (as nitrite) production was also determined as mentioned previously.

Immunization and sample collection

Three independent experiments were designed. Ovalbumin (OVA), (Sigma-Aldrich Inc., Saint Louis, MO, USA) was used as antigen (10 mg per animal in 50 µL of saline solution). All immunizations were subcutaneously applied (s.c.) in a total volume of 100 µL per animal (50 µL of miniemulsion and 50 µL of antigen). Freund’s Adjuvant Complete (FCA; Sigma-Aldrich Inc., Saint Louis, MO, USA) was used as positive control; a group of only antigens was used as control (ovalbumin 10 mg per dose in 100 µL of saline solution).

Experiment A. Mice inoculated with CN, CNZ, FCA and control were sacrificed by cervical dislocation at 24 h post-treatment. The administration site (subcutaneous tissue over the femoral area) was fixed in 10% formaldehyde solution and after that, covered with hematoxylin and eosin (H & E) to determine leukocyte infiltration. For the quantification and cell typing, mice were treated intraperitoneally with miniemulsions without antigen. The cells (cells/µL) were located by size and granularity to stablish the gate, and the CD14+ expression was determined by flow cytometry. These assays were carried out by flow cytometry (Accuri C6; DB Biosciences, San Jose, CA, USA).

Experiment B. Mice were immunized with OVA, and its combination with CN, CNZ, and FCA or control (saline solution) on days 0 and 14. Blood samples were taken on days 7, 14, 28, and 42. The blood samples were obtained from cardiac punction in anesthetized mice. Blood serum was separated and frozen at −80 °C until the time of isotype antibody determination (IgG1, IgG2a, IgG2b, IgG3, IgA and IgM) and determination of cytokine profile TH1 (IFN-γ, IL-2 and TNF-α)/TH2 (IL-4, IL-6 and IL-10)/TH17 (IL 71a). Cytokines were measured by flow cytometry (Accuri C6; DB Biosciences, San Jose, CA, USA) using a commercial Cytometric Bead Array (CBA) Mouse Th1/Th2/Th17 Cytokine kit (Catalog No. 560485; DB Biosciences, San Jose, CA, USA).

Experiment C. Mice were immunized at days 0 and 14 with the different miniemulsions (CN or CNZ) previously mentioned and challenged with OVA at day 42 and, blood samples were taken at 72 h after. The blood serum was separated to evaluate specific antibodies and frozen at −80 °C until analysis.

ELISA assay

Anti-ovalbumin antibodies were evaluated by ELISA, as previously described (Thommen Maciel Sartor, Moleta Colodel & Albuquerque, 2011). Briefly, polystyrene 96 well-plates (Falcon) were coated overnight at 4 °C with 0.5 mg of OVA per well in carbonate-bicarbonate buffer pH 9.6, after that washed with PBS 0.05%-Tween-20, and then blocked with 1%-BSA/PBS for 1 h at room temperature. Mouse antiserum from day 45 was added by triplicate (serial dilutions from 1:100) and incubated overnight at 4 °C. Plates were washed and incubated for 1 h at 37 °C with goat anti-mouse IgG-HRP (Catalog No. sc-2005; Santa Cruz Biotechnology Inc., Dallas, TX, USA). After that wells were washed, and tetramethylbenzidine (TMB) was added. The reaction was stopped with H2SO4. Plate was read at 450 nm in a plate reader (Synergy HT; Biotec, Emmerich, Germany), values indicating optical density (DO). The antibodies isotypes (IgG1, IgG2a, IgG2b, IgG3, IgA and IgM) were determinate by commercial kit Ig Isotyping Mouse Uncoated ELISA (Catalog No. 88-50630; Thermo Fisher Scientific Inc., Vienna, Austria).

Statistical analysis

All data analysis was performed with Minitab® (2014) software. Data was expressed as the mean ± standard error (SE). For the variables of nitric oxide production and antioxidant capacity, a Pearson correlation test was performed. Statistical significance (α = 0.05) was determined by ANOVA. The comparison of means was performed by Tukey test.

Results

Miniemulsion development and characterization

The miniemulsion CN and CNZ presented stability for 90 days, and this was confirmed by DLS analysis (Table 1 and Figs. 1A and 1B). Immediately after preparation, the CN miniemulsion presented a higher particle size (543.1 nm) with a zeta potential = −26.6 mV and PDI = 0.074, in contrast to CNZ that presented a size of 320 nm, a zeta potential = −28.1 mV and PDI = 0.449. However, after 90 days slightly differences were observed, the CN emulsion presented a decrease in zeta potential (−13 mV) and increase of PDI (0.225), and CNZ presented a decrease in size (259.4 nm) and PDI (0.312) but increase of zeta potential (−29.5 mV). The particle size difference of CN and CNZ was confirmed by atomic force microscopy (AFM) (Fig. 1C), where CN showed big semispherical structures and CNZ small spherical structures.

Table 1 DLS analysis of the miniemulsions CN and CNZ at 25 °C.

Miniemulsions	Time (months)	
0	1	3	
CN				
Droplet size (d.nm)	543.1	506.1	542.7	
PDI	0.074	0.192	0.225	
Zeta potential (mV)	−26.6	−22.1	−13	
CNZ				
Droplet size (d.nm)	320	266.1	259.6	
PDI	0.449	0.279	0.312	
Zeta potential (mV)	−28.1	−30.6	−29.5	

Figure 1 Particle size distribution measurement and AFM images of miniemulsions.

(A) Measurements of (1) CN and (2) CNZ by dynamic light scattering. (B) Measurements of zeta potential of (1) CN and (2) CNZ by dynamic light scattering. (C) Images by AFM 2D of (1) CN and (3) CNZ, and 3D of (2) CN and (4) CNZ.

The FT-IR spectra of ZnSO4 • H2O and ZnO nanoparticles is shown in Figs. 2A and 2B, respectively. In Fig. 2A, can be observed the characteristic peaks of ZnSO4 • H2O, a hydroxyl group (OH) at 3,135 cm−1, a carbonyl group (C=O) at 1,506 cm−1, and sulfate groups (SO−24) at 1,090 y 855 cm−1. The spectra of ZnO nanoparticles (Fig. 2B) showed a peak at 576 cm−1, characteristic of the ZnO stretching mode. Figure 2C indicates the principal groups of ZnSO4 • H2O and ZnO nanoparticles observed in FT-IR spectra. The photography by scanning electron microscopy (SEM) (Fig. 2C) showed agglomerates of ZnO nanoparticles with spherical morphology.

Figure 2 Characterization of zinc oxide nanoparticles synthesized.

(A) FTIR spectra of zinc sulfate (precursor). (B) FTIR spectra ZnO nanoparticles. (C) Scanning electron microscopy (SEM) image of the synthesized zinc oxide nanoparticles. (D) The X-ray diffraction picks observed correlated with JCPDS card No 01-089-0510 indicates the hexagonal phase of ZnO.

There is a qualitative difference between two samples in the size of the emulsions, considering that in Fig. 1 the emulsions are larger compared to Fig. 2, in addition a homogeneous size is observed in the case of the emulsions with CNZ, evidence of a large amount of residues in the first sample probably due to the inefficiency of the synthesis system, which indicates that in CNZ there is an improvement in the synthesis by adding zinc nanoparticles, probably due to its hydrophilic characteristics.

The Fig. 3 showed the FT-IR spectra of miniemulsions and their precursors. In cottonseed oil (Fig. 3A) was observed the C-H group strongly stretched at 2,922 and 1,465 cm−1; stretching vibrations of carboxyl group (COOH) at 2,856 cm−1, ketone group (C=O) at 1,744 cm−1, and C-O group at 1,156 cm−1. The spectra of α-tocopherol (Fig. 3B) showed peaks at 2,927 and 2,867 cm−1 that correspond to the asymmetric and symmetric stretching vibrations of CH2 and CH3, respectively. The peak was observed at 1,758 cm−1 for an alkene group (C=C), at 1,461 cm−1 for an asymmetric bending of a phenyl group (C6H5), and methyl and methylene symmetric bending peaks at 1,365 and 1,203 cm−1, respectively.

Figure 3 Characterization of miniemulsion CN and CNZ.

(A) FTIR spectra of cottonseed oil, (B) FTIR spectra of α-tocopherol, (C) FTIR spectra of CN, (D) FTIR spectra of CNZ and (E) FTIR spectra of miniemulsions CN, CNZ, and cottonseed oil.

The miniemulsion CN FT-IR spectra showed interaction of functional groups COOH, C=O and C-O of cottonseed oil with the C=C and C6H5 functional groups of α-tocopherol (Fig. 3C) in addition to the interaction of ZnO nanoparticles in CNZ miniemulsion (Fig. 3D).

Figure 3E represent the FT-IR spectra comparative of miniemulsions CN, CNZ, and cottonseed oil, and Table 2 showed the principal functional groups of precursors.

Table 2 Functional groups of cottonseed oil, α-tocopherol, and miniemulsion CN and CNZ.

FT-IR spectrum of miniemulsions	
Functional group	Wave number (cm−1)	
Cottonseed oil	
C-H	2,922	
COOH	2,856	
C=O	1,744	
C-H	1,465	
C-O	1,156	
α-Tocopherol	
CH2	2,927	
CH3	2,867	
C=C	1,758	
C6H5	1,461	
CH3	1,365	
CH2	1,203	
CN	
Interaction with the group COOH (cottonseed oil)	2,856	
Interaction with the group C=O (cottonseed oil)	1,745	
Interaction with the group C-O (cottonseed oil)	1,156	
Interaction with the group C=C (α-Tocopherol)	1,758	
Interaction with the group C6H5 (α-Tocopherol)	1,461	
CNZ	
Interaction with the group COOH (cottonseed oil)	2,856	
Interaction with the group C=C (α-Tocopherol)	1,758	
Interaction with the group C=O (cottonseed oil)	1,745	
Interaction with the group C6H5 (α-Tocopherol)	1,461	
Interaction with the group C-O (cottonseed oil)	1,156	
Nanoparticles of ZnO	670	

Effect of miniemulsions on immune cell function

CN and CNZ miniemulsions at concentrations of 1.25 to 10 µL/mL had no effect on the viability of macrophages (p = 0.108), splenocytes (p = 0.413), and thymocytes (p = 0.923) in vitro (Fig. 4). All doses tested of CN and CNZ induced nitric oxide production in dose dependent manner when compared with the control (0 µL). The CN induced a similar production of nitric oxide at doses of 1.25 (p = 0.998) and 2.5 (p = 0.112) µL/mL and highest at doses of 5 and 10 µL/mL (p = 0.0001) when compared with the positive control (LPS). No difference (p = 0.935) in nitric oxide production was found at all doses tested of CNZ when compared with LPS (Fig. 5).

Figure 4 Effect of the miniemulsions on cell viability.

The miniemulsions (CN and CNZ) at concentrations of 1.25 to 10 µL/mL did not affect the viability of macrophages, splenocytes and thymocytes (2 × 105 cells/well with three replicates) evaluated by MTT. Values are expressed as mean ± SE. The analysis was performed by means of an ANOVA test with a significance of p < 0.05.

Figure 5 Production of nitric oxide in murine peritoneal macrophages.

Murine peritoneal macrophages (2 × 105 cells/well with three replicates) were stimulated with different miniemulsions (CN and CNZ) doses. LPS 5 µg/mL was used as control (+) and cells without treatment as control (−). The nitric oxide were determinate as nitrite. An ANOVA (p < 0.05) was performed to compare the effect of the different doses. To determine the difference between treatments, a comparison of means was performed by Tukey.

The antioxidant capacity

All CN and CNZ doses tested had the capacity to induce the production of antioxidants by dose dependent manner when compared with the control (p = 0.0001). The CN 5–10 µL/mL doses induced a major antioxidant production (p = 0.0001) than LPS and 2.5 µL/mL (p = 0.979)/1.25 µL/mL (p = 1.0) doses induced similar productions levels. The CNZ treatment at 10 µL/mL (p = 0.002) dose induced a higher antioxidant total production than LPS, but at minor doses 5 µL/mL (p = 0.115), 2.5 µL/mL (p = 0.925) and 1.25 µL/mL (p = 1.0) no difference was found (Fig. 6). There is a strong correlation between the production of nitric oxide and antioxidants in macrophages treated with CN (r = 0.905, p = 0.0001) as in those treated with CNZ (r = 0.804, p = 0.002).

Figure 6 Total antioxidant capacity (TAC) in murine peritoneal macrophages.

Murine peritoneal macrophages (2 × 105 cells/well with three replicates) stimulated with different doses of miniemulsions (CN and CNZ). LPS 5 µg/ml was used as control (+) and cells without treatment as control (−). An ANOVA (p < 0.05) was performed to compare the effect of the different doses. To determine the difference between treatments, a comparison of means was performed by Tukey.

Histological evaluation

The histological evaluation of the tissue at the inoculation site (subcutaneous tissue on femoral area) with CN or CNZ with ovalbumin showed inflammation with higher leukocyte infiltration compared with CFA (Fig. 7). The intraperitoneal inoculation with CN, CNZ miniemulsion and CFA showed a high recruitment of total intraperitoneal cells with the CD14+ phenotype, after 24 h of inoculation compared with control (p = 0.0001), however no difference was found between CN, CNZ and FCA (Fig. 8).

Figure 7 Leukocytes migration to the inoculation site of the miniemulsions.

H & E-stained histological images at 40X of the s.c. inoculation site of treatments (A) control without adjuvant, (B) ACF, (C) CN and (D) CNZ using Ovalbumin as antigen. Inflammation and leukocyte infiltration (black arrow) are observed in treatments (B) ACF, (C) CN and (D) CNZ.

Figure 8 Quantification of macrophages recruited to the inoculation site.

Intraperitoneal macrophages (n = 12 mices, three per tratment) recruited were determinate by flow cytometry. (A) Gate selection (macrophage population) by size and granularity, and reading by the FL1 channel of the CD14+ receptor. (B) Estimation of the amount (macrophages/µL) by size and granularity in murine peritoneal macrophages stimulated in situ (in peritoneum). (C) CD14+ expression in murine peritoneal macrophages stimulated in situ for 24 h. All statistical analysis were performed by means of an ANOVA, determining differences (p < 0.05) by means of a Tukey analysis.

Macrophage uptake and phagocytosis

The CN (p = 0.158) and CNZ (p = 0.495) miniemulsions evaluated by neutral red assay did not affect the macrophages uptake activity when compared with untreated macrophages (Fig. 9). However, in the macrophages Candida albicans-phagocytosis assay the CN and CNZ miniemulsions increased the phagocyte capacity with respect to the untreated macrophages (Fig. 10). Next, the evaluation of residual colony forming units indicated that only CN significantly decreased (p = 0.004) this value respect CNZ, but at 4 h no difference was found with respect to the untreated macrophages (Fig. 11A). By other side, only CN miniemulsion increased (p = 0.002) the nitric oxide production in all times evaluated (Fig. 11B).

Figure 9 Uptake activity of macrophages stimulated with the miniemulsions.

Macrophages (2 × 105 cells/well with seven replicates) were stimulate with the miniemulsions CN, CNZ or negative control for 2 h. Disodium clodronate (0.6 mg/mL) was used as non-phagocytic (Control −) (2 × 105 cells/well with four replicates). An ANOVA (p < 0.05) was performed to compare the effect of the different treatments. To determine the difference between treatments, a comparison of means was made by Tukey.

Figure 10 Qualitative evaluation of macrophage phagocytic activity.

The phagocytic activity was evaluated by means of 100X fluorescence microscopy in macrophages treated with (A) Clodronate disodium (0.6 mg/mL) as (Control −) of no phagocytosis, (B) without treatment, (C) CN (10 µL/mL) and (D) CNZ (10 µL/mL).

Figure 11 Phagocytic activity of macrophages co-cultured with C. albicans.

The co-cultured (n = 24) macrophages (1 × 106 cells)/C. albicans (1 × 106–1 × 108 cells) were treated with miniemulsions (CN or CNZ) at four times. (A) Residual colony forming units (CFU) after different times in co-culture. (B) production of nitrite oxide (NO) measured in the form of nitrite induced by the co-cultivation. The values are expressed as the mean (± SE). An ANOVA (*p < 0.05) was performed to compare the effect of the different doses. To determine the difference between treatments, a comparison of means was made by Tukey.

Cytokine production

The CNZ miniemulsion stimulated a major IFNγ secretion compared with FCA at day 7 (p = 0.0001), this difference was not observed at day 14 (p = 1.0) (Fig. 12). A major secretion of IL-2 was observed at days 7 (p = 0.027) and 14 (p = 0.007), stimulated with the CN and CNZ miniemulsions (Fig. 12). However, the IL-10 was not affected (Fig. 12).

Figure 12 Effect of miniemulsions on cytokine production in a murine model.

Serum levels (Pg/mL) of cytokines (IFNγ, IL2 and IL 10) in BALB/c mice (n = 30) inoculated s.c. with 50 µl of miniemulsions (CN and CNZ), positive control FCA or negative control an antigen (ovalbumin without adjuvant) on days 0 and 14 of the protocol. At 7, 14 and 28 days after the first inoculation. Ovalbumin (OVA) was used as antigen in all treatments. Values are expressed as mean ± SE. When a difference was observed (p < 0.05), a Tukey test was performed to determine differences.

Switching isotype evaluation

The CN and CNZ miniemulsions did not affect the production of IgG1. However, the FCA stimulated a major production of IgG1 at day 42 compared with the untreated mice (p = 0.044) (Fig. 13). The CN and CNZ miniemulsions did not affect the production of IgG2b at all times evaluated, only significant difference (p = 0.044) was found with FCA at day 28 (Fig. 13). At the last, the miniemulsions did not affect the production of antibody isotypes (IgG1, IgG2a, IgG3, IgA and IgM) at days 7, 14, 28 and 42.

Figure 13 Effect of miniemulsions on antibody isotypes production in a murine model.

Antibody isotypes (IgG1, IgG2a, IgG2b, IgG3, IgA and IgM serum levels) in BALB/c mice (n = 30) inoculated s.c. with 50 µl of miniemulsions (CN and CNZ), positive control (FCA) or negative control (ovalbumin without adjuvant) on days 0 and 14 of the protocol. Ovalbumin (OVA) was used as antigen in all treatments. Values are expressed as mean ± SE. When a difference was observed (*p < 0.05), a Tukey test was performed to determine differences.

IgG specific OVA

The CN miniemulsion induced a significant production of IgG against OVA, but lesser than FCA. CNZ miniemulsion did not affect this, when compared with the OVA treatment (p = 0.0001) (Fig. 14).

Figure 14 Effect of miniemulsions on specific antibodies production in a murine model.

Specific antibodies against OVA (IgG) 45 days after the last immunization (antigen + adjuvant). Mice (n = 30) were challenged with the antigen (OVA) on day 42 post vaccination. Values are expressed as mean ± SE. When a difference was observed (p < 0.05), a Tukey test was performed to determine differences.

Discussion

The droplet size obtained from CN (543.1 nm) and CNZ (320 nm) formulations, the zeta potential (−26.6 mV for CN and −28.1 mV for CNZ) and PDI values (0.074 for CN and 0.449 for CNZ) indicate a monodisperse emulsion with a greater molecule repulsive strength which increases its stability (Malik, Ameta & Singh, 2016).

The results of FT-IR spectra confirmed the synthesis of ZnO nanoparticles with a peak at 576 cm−1, characteristic of the ZnO stretching mode (Del Buono et al., 2022; Nagaraju et al., 2017), and the images by SEM showed agglomerates with a spherical morphology (Vasile et al., 2015; Shah et al., 2016). Similarly, FT-IR spectra confirmed the nature of cottonseed oil precursors (carboxyl at 2,856 cm−1, and ketone at 1,744 cm−1) (Bajaj, Tekade & Manoja, 2015), and α-tocopherol (alkene at 1,758 cm−1, phenyl at 1,461 cm−1, methyl at 1,365 cm−1, and methylene at 1,203 cm−1 groups) (Lucarini et al., 2020).

The charges detected by Z potential measurements can be attributed to the unsaturated fatty acids group of cottonseed oil, these molecules have a negative charge (El-Mallah et al., 2011; Shah et al., 2017). In fact, Tween 20 is a non-ionic surfactant, but is part of the emulsion, the charge depends on the interaction of phases (Xin et al., 2013).

The miniemulsion with ZnO nanoparticles showed more stability (ζ potential values CN = −13 mV CNZ = −29.5 mV at 3 months) due to the interaction of solid particles with the oil/water interface, this effect increases the stability by steric barrier, this is known as a Pickering stabilizer (Yang et al., 2017).

In our study the stimulation with CN and CNZ miniemulsions could activate the nitric oxide production in macrophages, indicating their activation. This activation is similar when macrophages are challenge with antigens, adjuvants, or vaccines, inducing nitric oxide production and the capacity to increase the immune response (Van den Biggelaar et al., 2020). The production of nitric oxide induced by the CNZ and CN miniemulsions offers interesting results, since reactive oxygen intermediates have a great influence on immune responses by activating or inhibiting the action of certain transcription factors or cytokines and by modifying pathways of programmed cell death (Arruda et al., 2004).

The antioxidant total production capacity was studied, determining that both miniemulsions have the capacity to induce higher levels of antioxidants, that can be correlated with the promotion of mitochondrial biogenesis increasing antioxidant enzymes production, decreasing the exacerbation of proinflammatory cytokine response, and blocking or scavenging the excessive response of ROS and RSN, that in higher amounts can affect a healthy immune response (Maldonado et al., 2021). Is important to mention that cottonseed oil contain antioxidants that can act as exogenous antioxidant sources, however, is possible that antioxidants could be synthesized through different intracellular mechanisms.

The results such as leukocyte infiltration after subcutaneous injection and the increase of intraperitoneal cells recruitment, predominantly CD14+ after intraperitoneal miniemulsions injection, similar to the MF59 adjuvant, that is an oil-in-water emulsion containing squalene (4.3%) in citric acid buffer with stabilizing non-ionic surfactants Tween 80 (0.5%) and Span 85 (0.5%) used in vaccines (Ko & Kang, 2018). Being the MF59 capable to create an immunocompetent environment at the injection site by recruiting immune cells like neutrophils, macrophages, granulocytes, monocytes, and other immune cells that ingest the antigen (Shah et al., 2015). Also, in macrophages the CN and CNZ promotes the phagocytic capacity, similarly to MF59, where antigens and MF59 are phagocyted by neutrophils and monocytes, and later by dendritic cells (DCs) and B cells, then moved to draining lymph nodes (Ko & Kang, 2018). The CN miniemulsion induced the nitric oxide production in mice, likely to FCA where high nitric oxide production is observed (Kahn et al., 2001). Nitric oxide is considered a dual molecule in relation to its benefit on immune system; at low concentration plays a healthy role on immune system but at higher levels immunosuppression is induced (Shreshtha et al., 2018). This study is not designed to determine the role of nitric oxide, however for the data obtained we assumed that CN or CNZ miniemulsion can activate healthy immune responses.

Several studies report that IFN-γ derived from innate immune cells enhances protective antiparasitic Th1 responses and malaria protection after use of adjuvant AS01 (containing two immunostimulants (TLR4 ligand 3-O-desacyl-4′-monophosphoryl lipid A (MPL) and the purified saponin QS-21) in a liposome-based formulation) or AS03 (containing α-tocopherol and squalene in an oil-in-water (o/w) emulsion). In addition, the production of IFN-γ could be mediated by IL-2 produced by vaccine-induced effector memory T cells (Burny et al., 2017). Increase in the IFN-γ and IL-2 production were found in our study with the CN or CNZ miniemulsions treatment.

Only CN miniemulsion induced a significant OVA specific IgG production, however no IgG isotype antibodies switch was observed. This can suggest the lack of CD4+ T cell dependency (Ko & Kang, 2018). However, to define these results is necessary the use of CN miniemulsion combined with OVA in CD4+ deficient mice, and to stablish the dependence of Th1 or Th2.

The detection of IgG at day 45 can suggest a sustained immunity induced by lymphocyte B and T using CN miniemulsion. The fact of finding a higher IgG concentration induced by FCA than CN miniemulsion can be attributed to the formation of FCA depots at the injection site, inducing delayed germinal center formation promoting higher antibody responses than a non-depot- inducing adjuvant. Such as MF59-like squalene emulsion AddaVaxTM after a single immunization (Pedersen et al., 2020). These results indicate that CN miniemulsion is a non-depot inducing adjuvant.

According to our results, we suggest that our miniemulsions have mechanisms of action similar to MF59-like squalene emulsion AddaVaxTM. Oils, such as olive, rice bran, corn, peanut, rapeseed, sunflower, and cottonseed contain 0.1 to 4 g/Kg of squalene (He & Corke, 2003; Kalogeropoulos & Andrikopoulos, 2004). Commercial adjuvants such as MF59 and AS03 contain 0.0975 and 0.1069 g per dose, respectively (Shah et al., 2015). If compared commercial squalene adjuvants doses with maximum content in vegetable oils is necessary to use 2.43–2.67 g of oil per dose. This dose is perfectly feasible for farm animals and humans. Vegetable oils can be used in alternative adjuvants to squalene-based emulsion for vaccination, because they contain a long variety of active substances that have effects over immune system (Fox et al., 2011).

Conclusions

In this study, we synthesized two new miniemulsions cottonseed oil-based, containing or not ZnO nanoparticles with adjuvant and antioxidant capacity, capable of generating leukocyte infiltration and increase cytokines and antibodies production. However, it is necessary to carry out evaluations to determine its effect on the immune system with various antigens and species.

Supplemental Information

Supplemental Information 1 Raw data of experiments.

ELISAs and flow cytometry.

Click here for additional data file.

We thank the MsC. Alejandra Arreola Triana for proofreading this article.

Additional Information and Declarations

Competing Interests

Author Contributions

Animal Ethics

Data Availability

The authors declare that they have no competing interests.

Gustavo Sobrevilla-Hernández conceived and designed the experiments, performed the experiments, analyzed the data, prepared figures and/or tables, authored or reviewed drafts of the article, and approved the final draft.

Moisés Armides Franco-Molina conceived and designed the experiments, analyzed the data, authored or reviewed drafts of the article, and approved the final draft.

Diana G. Zárate-Triviño conceived and designed the experiments, authored or reviewed drafts of the article, and approved the final draft.

Jorge R. Kawas conceived and designed the experiments, authored or reviewed drafts of the article, and approved the final draft.

Sara Paola Hernández-Martínez performed the experiments, analyzed the data, prepared figures and/or tables, and approved the final draft.

Paola Leonor García-Coronado performed the experiments, prepared figures and/or tables, authored or reviewed drafts of the article, and approved the final draft.

Silvia Elena Santana-Krímskaya analyzed the data, prepared figures and/or tables, and approved the final draft.

Cynthia Aracely Alvizo-Báez analyzed the data, prepared figures and/or tables, and approved the final draft.

Cristina Rodríguez-Padilla analyzed the data, prepared figures and/or tables, and approved the final draft.

The following information was supplied relating to ethical approvals (i.e., approving body and any reference numbers):

CEIBA, Laboratorio de Inmunología y Virología, Facultad de Ciencias Biológicas, Universidad Autónoma de Nuevo León(ceiba-2019-015).

The following information was supplied regarding data availability:

The raw data is available in the Supplemental Files.

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
