# Peer review of "Development of a new generation of miniemulsion based on cottonseed oil with α-tocopherol and ZnO and evaluation of its adjuvant activity"

_PeerJ, doi:10.7717/peerj.14981_

## Round 0.1 · original submission · Major Revisions

There are a number of major concerns raised by review. These are primarily listed under 'validity of the findings'. I am minded to agree with most of these points and would therefore ask you to provide a full and detailed explanation and assessment of these issues in revision. Information about sizes is important and must be provided.

I also ask that you increase the detail and depth of your figure legends to explain how many technical and biological replicates were performed for all biological assays (e.g. - but not restricted to - Figures, 4, 5, 6 etc) and to provide all raw data for all replicates of all experiments, (e.g., raw data for FACS etc). There is insufficient primary data provided for value. This is essential.

Please address all these points in your rebuttal if you are willing to perform the additional work required.

Reviewer 1 ·

Basic reporting

In the present manuscript, the authors study the adjuvant activity of oil-in-DMSO emulsions with alpha-tocopherol and cottonseed oil in the disperse phase. They compare the activity of emulsions with and without ZnO nanoparticles, labeled as CNZ and CN, respectively.

* LANGUAGE AND STYLE. There are a few language and expression problems along the text. The authors should revise carefully the text, perhaps with the assistance a colleague proficient in English. Some examples are the following:
a) Line 67: subject missing in the sentence (“Can be derived…”); in English the subject cannot be omitted.
b) Line 71-72: “Different emulsion types have been used as immunostimulants or adjuvants in therapeutic or prophylactic vaccination; which are formed when two immiscible liquids” --> problem with the relative clause (and with the punctuation) --> The relative clause introduced by WHICH refers to EMULSIONS; it cannot be placed in this position (and it should also not be preceded by semicolon).

* USE OF THE TERM “NANOEMULSION”. Another important terminological aspect is the use that the authors make about the term NANOEMULSION. Assuming the droplet sizes given by the authors in Table 1 as correct, they are beyond the size generally considered as “nanoemulsions” (below 100 nm, up to 200 nm at most). The systems may be considered perhaps as “MINIEMULSIONS”, but “nanoemulsion” does not seem to me as a proper name. In case of doubt, the general term “emulsion” should be used.

* FIGURES
a) Figure 2: I would avoid the combination of tables with figures. In addition, panels A and B are too small to be properly seen.

Experimental design

The design of the experiments is correct. However, there are a few points in the experimental section to be taken into account:
a) All reactants should be included in the “reactants” section, also NaOH, Tween 20, and DMSO.
b) Line 124: “zeta potential” is not determined by dynamic light scattering, at least not directly. In the instrument used by the authors, zeta potential is determined from electrophoretic mobility measurements. The electrophoretic mobility is determined from scattering measurements, but it is not correct to state that “zeta potential is determined by DLS”.

Validity of the findings

Although the paper focuses on the biomedical application of the emulsions, which seem to be the main expertise of the authors, there are important points about colloid science that cannot be neglected and should be improved:
a) The values obtained by DLS are zeta potential are doubtful. For the sample CN, the size does not seem to change after 3 months. It is not clear why the values are lower after 1 month. Taking into account that the surfactant used is a non-ionic one (Tween 20), the changes in zeta potential are not clear (they should be in principle close to zero, as least for the samples without ZnO). The negative values are probably related to the ions adsorbed in the electric double layer of the droplets.
b) The AFM results should be taken very carefully. I really doubt that AFM is able to observe anything else than the
c) It would be convenient to have more information about the particle size of the ZnO nanoparticles, either from DLS (if a reasonable dispersion can be obtained) or from transmission electron microcopy (TEM) images. It would be also important to know about the aggregation of the nanoparticles.
d) Is it possible that the ZnO is acting as a Pickering stabilizer in the emulsions with CNZ sample (i.e., stabilization of the droplets by the ZnO nanoparticles)? The authors could probably explore and comment on this possibility.

Additional comments

A few terminology aspects to revise:
a) Abstract: “were characterized by microscopy images…” --> “were characterized by electron microscopy”, write the technique (microscopy), not the outcome (images)
b) Line 73: “Emulsions can have a formula of water-in-oil…” --> rephrase in a more correct way, “formula” is not correctly used here. Write, for instance, “Emulsion can be formulated either as water-in-oil or oil-in-water” (or something similar).
c) Line 96: “zinc in nanoparticle form” --> wrong from a chemical point of view, you could have “zinc nanoparticles” (metal nanoparticles), but it is not what the authors refer here, so it should be phrased in appropriate, chemically correct terms.
d) Line 127: “electron microscopy” instead of “electron microscope”
e) Line 127: “the nanoemulsion size was determined by AFM” --> not correct. If at all, the droplet size, but not the “nanoemulsion size”. However, in any case, I find quite doubtful to determine droplet sizes with AFM

Overall, the paper is interesting and may be worthy of publication. However, in my opinion, the authors should revise and take into account the points indicated above before the paper can be accepted.

---

## Round 0.2 · Minor Revisions

Could you address the remaining point?

Thanks for carefully attending to all other points raised.

Reviewer 1 ·

Basic reporting

- For future occasions, I would very strongly recommend the authors to include the comments or questions of the reviewers/editor before each answer. Otherwise, it is quite tedious to identify what has been answered and what not.
- The authors have revised the language, following the recommendation.
- Regarding the term "nanoemulsion", I disagree with the answers provided. The fact that some authors use it wrongly in JCR publications, does not justify a further wrong use. Technically speaking, emulsions above 100 nm (or 200 nm at most) are clearly NOT NANOEMULSIONS. However, you can perfectly use the term MINIEMULSION, which is perfectly fine for the proposed range (50 to 600 nm).
- Figrure 2 has been corrected.

Experimental design

- The requested aspects have been addressed.

Validity of the findings

- The requested aspects have been addressed.
- Regarding DLS and zeta potential results, I am personally not completely convinced about the given arguments, but I appreciate the attempt of the authors to explain my concerns.

Additional comments

The additional comments were also addressed.
My only change request regards the term "nanoemulsion".

---

## Round 0.3 · accepted · Accept

Thanks for this. Congratulations!